# DRIFT TYPE AND MAGNITUDE DETECTION IN IMAGE CLASSIFICATION NEURAL NETWORKS

## ABSTRACT

A change in the input data stream of a machine-learning model is referred to as a data drift and may impact the model's accuracy. This paper proposes a framework to detect data drifts, identify the type of drift, and estimate the drift magnitude that occur in the input data stream of image classification neural networks due to various effects. It applies to any type of drift that occurs in images due to various factors such as noise, weather, etc. A novel statistical method is proposed for drift magnitude estimation. The method relies on the change in the prediction probability distributions of the predicted classes in the classification network caused by the data drift. The drift magnitude is estimated by applying a set of thresholds to the prediction probabilities. The drift type is identified using a classification neural network. Experimental results obtained using various datasets, drift types, and neural network architectures show that the proposed framework can accurately detect data drifts, accurately identify the drift type, and estimate the drift magnitude with a very low quantization error.

## 1 INTRODUCTION

As a result of the rapid advancement in artificial intelligence and computing power, the adoption of machine intelligence in modern applications is continually growing. Computer vision stands out as one of the prominent areas where machine intelligence is widely applied. Image classification is a fundamental computer vision task accomplished through machine learning techniques. There are advanced Deep Neural Network (DNN) models for image classification that have achieved remarkable success, such as AlexNet (Krizhevsky et al. (2012)), VGG (Simonyan & Zisserman (2014)), GoogleNet (Szegedy et al. (2015)), and ResNet (He et al. (2016)). Utilization of image classification DNNs has become more prevalent across a broad range of applications. For instance, self-driving cars (Ni et al. (2020)), mobile computing devices (Zhang et al. (2018)), medical imaging (Sarvamangala & Kulkarni (2022)), surveillance (Kaljahi et al. (2019)), defect detection (Czimmermann et al. (2020)), etc.

Machine learning models such as DNNs are trained using data-driven statistical learning schemes. During the operation time, a model is expected to exhibit similar accuracy as in the training phase assuming a static distribution of the input data. However, the real-world application environments are profoundly dynamic and continuously changing. The assumption of a static distribution of the data is not valid in many of the applications, making it challenging for the models to maintain a high accuracy during their operation. A change in the distribution of the input data that impacts the accuracy of a machine learning model is referred to as data drift. Such data drifts occur over time due to various reasons and the model accuracy may drop below the safety margin.

Image data streams are highly susceptible to data drifts due to noise effects (such as Gaussian, Poisson, Salt & Pepper, Speckle), weather effects (such as Snow, Fog, Rain, Shadow), degradation of the camera, change in lighting, etc. Such data drifts impact the accuracy and may compromise the system reliability. The accuracy degradation of the image classification neural networks due to data drifts is discussed by Hashmani et al. (2019). Diagnosing data drifts occur in image classification neural networks is an essential aspect of maintaining the system reliability.

Detection and magnitude estimation are two main aspects of the drift diagnosis process (Webb et al. (2016), Webb et al. (2017)). There exist many approaches for the detection of data drifts in various application domains (Yang et al. (2020), Suprem et al. (2020), Dube & Farchi (2020), Ackerman

et al. (2021), Senarathna et al. (2023)). It is important to know the drift magnitude for tasks such as determining the detection threshold, appropriate resource allocation, model performance assessment, model adaptation, risk management, etc. The detection and magnitude estimation of drifts in one-dimensional data can be done by computing the distance between the distribution of the training data and the production data (Goldenberg & Webb (2019)). Various distance matrices are used to identify data drifts in one-dimensional data. For instance, Kullback-Leibler Divergence (Kullback & Leibler (1951)) and Hellinger Distance (Hoens et al. (2011)) are used by Webb et al. (2016), Total Variation Distance (Levin & Peres (2017)) is used by Webb et al. (2017), and Energy Distance (Rizzo & Székely (2016)) is used by Yang et al. (2020). However, these matrices cannot be directly applied to higher dimensional data with spatial correlations, such as images.

There are methods in the literature for the detection of data drifts in image data streams (Suprem et al. (2020), Dube & Farchi (2020), Ackerman et al. (2021)). Some methods use a low-dimensional feature vector extracted from the image to address the challenge of processing the high-dimensional image data (Suprem et al. (2020)). The method by Suprem et al. (2020) maps the images into a low-dimensional feature vector using a Generative Adversarial Network (GAN) and detects data drifts by applying clustering on the low-dimensional feature vector. The method by Dube & Farchi (2020) uses the divergence score between the average feature vectors of the training dataset and the production dataset to detect data drifts in image classification neural networks. Ackerman et al. (2021) proposed a method in to detect data drifts using the prediction probability i.e. the maximum class probability. The change in the prediction probability distribution resulting from the data drift was employed to detect a data drift by Ackerman et al. (2021). However, the methods by Suprem et al. (2020), Dube & Farchi (2020), Ackerman et al. (2021) do not estimate the drift magnitude, and they only consider the drifts due to the appearance of unseen classes in classification neural networks. There exist methods that estimate the magnitude of various noise effects in images (Pyatykh et al. (2012), Chuah et al. (2017), Wang et al. (2021)). However, those methods rely on the statistical properties of the noise model, and therefore they only apply to a particular noise type.

The method proposed by Senarathna et al. (2023) also relies on the change in the prediction probability distribution due to the data drift and estimates the drift magnitude by applying thresholds to the prediction probabilities. However, Senarathna et al. (2023) assumed that the distribution of the images among the classes, referred to as the class distribution, remains unchanged in training and production data. This assumption often does not hold in real-world application environments. In this paper, we address this limitation and propose a method that can cope with varying class distributions.

To address the problem of varying class distribution of the images in the input data stream, we combine the thresholding criterion proposed by Senarathna et al. (2023) with an approach that estimates the number of images from each class in the input data stream. The process of estimating the number of samples from each class in an input data stream of a machine learning model is referred to as Quantification (Forman (2005)). With that, we derive a generalized method to estimate the magnitude of data drifts due to any type of effect under varying class distributions. The proposed method relies on the change in the prediction probabilities of the neural network caused by the data drift. It is a statistical method that operates on a batch of images and estimates a discrete magnitude for the batch. Using the proposed drift magnitude estimation method, we developed a comprehensive drift detection framework that detect data drifts, identify the type of drift, and estimate the drift magnitude in the input data stream of image classification neural networks due to various effects. Drift type is identified using a secondary classification neural network that runs in parallel with the primary classification network. The drift type is detected from a set of potential drift types and the drift magnitude is estimated from a set of potential drift magnitudes.

The rest of the paper is organized as follows. Section 2 presents preliminaries and Section 3 describes the proposed method. Section 4 discusses the experimental results and conclusions are presented in Section 5.

## 2 PRELIMINARIES

Changes in the input data distribution of a machine learning model are called concept drifts. A concept drift is a change that occurs in the relationship between the input features and the target variable of a machine learning model over time in an unforeseen manner. Concept drifts are categorized as

real concept drifts, covariate drifts, and novel class appearance (Hoens et al. (2012), Gama et al. (2014), Webb et al. (2016), Webb et al. (2017)). A real concept drift, which is also referred to as class drift or prior probability shift, is a change in the relation between the input and output. A covariate drift, which is also referred to as virtual drift, is a change in the input data distribution. The novel class appearance is a previously unseen new class appears in the input. This paper specifically focuses on covariate drifts that occur in image classification neural networks due to various effects such as noise effects and weather effects among others. Furthermore, we consider "full concept drifts" where every input in the input data stream is equally affected with the same drift magnitude (Webb et al. (2016)).

To understand the impact of data drifts on image classification neural networks, let us consider the impact of Gaussian noise on a classification network trained on the MNIST handwritten digit image dataset, detailed in Section 4, under the effect of Gaussian noise. Drift magnitude is represented by the variance $\sigma^2$ of the Gaussian noise. The network had an accuracy of 99.10% on clean images i.e. $\sigma^2 = 0$. Accuracy gradually declined as the drift magnitude increased and dropped below 80% when the images were affected by the Gaussian noise of $\sigma^2 = 0.20$. See also Figure A.1 in the Appendix.

Softmax is the activation function used in the output layer of classification neural networks. The softmax activation provides a normalized distribution of probabilities among the output classes and the predicted class for a given input is the class that has the maximum softmax probability. The maximum softmax class probability is referred to as the prediction probability and it is an indication of the confidence of the prediction made by the classifier. The distribution of the prediction probabilities changes when a data drift occurs. Hendrycks & Gimpel (2016) showed that the average prediction probability decreases when out-of-distribution data is fed into a neural network classifier.

Let us consider the above mentioned MNIST classifier. Figure 1 shows the Cumulative Distribution Functions (CDFs) of the prediction probability of the images predicted as class 1 and class 2 by the MNIST classifier under six different magnitudes of Gaussian noise. The distribution of the prediction probability changes at different magnitudes of Gaussian noise. Furthermore, the prediction probability distributions of different predicted classes change differently with the drift magnitude. For instance, the change in the prediction probability distribution of class 0 in Figure 1a and class 1 in Figure 1b are different. The thresholding-based criterion proposed by Senarathna et al. (2023) to estimate the magnitude of data drifts in image classification neural networks relies on this change in the prediction probability distribution.

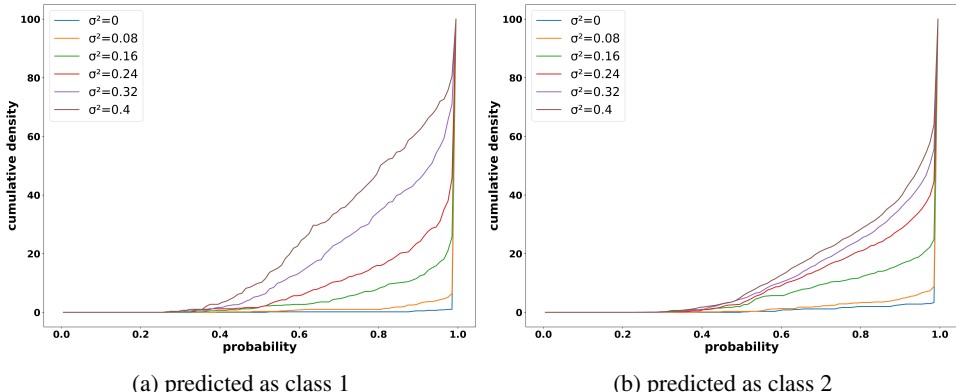

(a) predicted as class 1        (b) predicted as class 2

Figure 1: Cumulative Distribution Functions (CDFs) of the prediction probability of the images predicted as class 0 and class 1 in the MNIST classifier under different magnitudes of Gaussian noise.

However, the distribution of the prediction probability of a given predicted class depends on the class distribution of the input data stream. Because, given that the predicted class by the classification network of an input is $C$, the actual class $\hat{C}$ of the input can be any class due to the model imperfections. Therefore, the number of images predicted as class $C$, and subsequently the distribution of the prediction probabilities of images predicted as class $C$ depends on the class distribution of the input data stream.

The relation between the prediction probability distribution and the class distribution is as follows. Let $P_i$ denote the prediction probability distribution of the images predicted as class $i$, $P_{i,j}$ denote the prediction probability distribution of the images belonging to the class $j$ predicted as class $i$, and $x_j$ denote the number of images from class $j$ for $1 \leq j \leq n$ in the input data stream where $n$ denotes the number of classes. Then,

$$P_i = \frac{\sum_{J=1}^{n} P(C = i | \hat{C} = j) x_j P_{i,j}}{\sum_{J=1}^{n} P(C = i | \hat{C} = j) x_j}, \qquad (1)$$

where $P(C = i | \hat{C} = j)$ denotes the conditional probability of being predicted as class $i$ given that the actual class is $j$. In other words, $P(C = i | \hat{C} = j)$ indicates the probability of the images belonging to the class $j$ being predicted as class $i$ where $1 \leq i, j \leq n$. To address the challenge of varying class distributions, we combine the thresholding criterion proposed by Senarathna et al. (2023) with a quantification method to estimate the drift magnitude under varying class distributions in the input data stream. There exist numerous quantification methods in the literature to estimate the number of samples from each class in multiclass classification networks (Schumacher et al. (2021)). The quantification method proposed in this paper is motivated by the *readme* method by Hopkins & King (2010).

## 3 THE PROPOSED METHOD

The proposed drift detection approach considers a batch of images at a time. A set of potential drift types and a set of potential discrete drift magnitudes per each drift type are considered. The approach detects the presence of a data drift, identifies the drift type (i.e. effect type) and estimates the drift magnitude for a batch of images. First, a drift magnitude is estimated for each potential drift type. Based on the estimated magnitudes for each type, it is determined whether a data drift has occurred or not. If a data drift has occurred, then the drift type is identified using the type detection criterion. The drift magnitude corresponds to the estimated magnitude of the detected drift type.

For a given drift type, the magnitude is estimated based on the percentage of images in each predicted class with a prediction probability above a predetermined threshold. The thresholds are computed using the method proposed by Senarathna et al. (2023). We first describe the threshold computation criterion and the application of the thresholds assuming a static class distribution in the image data stream.

The percentage of images with prediction probability higher than an appropriately selected threshold differs at different drift magnitudes. Consider the CDFs shown in Figure 1a of the prediction probabilities of the images predicted as class 1 in the MNIST classifier. The noticeable difference in the cumulative sum at different drift magnitudes indicates the possibility to distinguish among the different drift magnitudes by choosing an appropriate threshold. For instance, let us consider the number of class 1 predictions with prediction probability $\geq 0.9$ at different Gaussian noise magnitudes in Figure 1a. The corresponding percentage for $\sigma^2 = 0$, $\sigma^2 = 0.04$, $\sigma^2 = 0.08$, and $\sigma^2 = 0.12$ were 99.3%, 95.4%, 86.0%, and 71.1% respectively. The drift level is recognized based on these differences in the percentages above some threshold probability.

Let $T$ denote the drift type and $M$ denote the drift magnitude. Let $m$ be the number of non-zero drift magnitudes. For a potential drift type $T$, multiple threshold probability values are computed. In particular, a different threshold is computed for each predicted class $C$, for each drift magnitude $M$ of the drift type $T$. Let us consider the CDFs shown in Figure 1. It can be observed that the probability that has the highest contrast among the CDFs of consecutive drift levels varies based on the drift magnitude and the predicted class. Therefore, a different threshold is computed per each predicted class $C$ and per each drift magnitude $M$ such that the difference among the percentages above the thresholds of consecutive drift levels is maximized. Hence, a better distinguishability among the adjacent drift levels is achieved.

Let $\tau_{C,M}$ denote the threshold probability for the predicted class $C$, drift magnitude $M$ where $1 \leq C \leq n$ and $0 \leq M \leq m$. For some drift type $T$, the thresholds $\tau_{C,M}$ for each predicted class $C$ and each drift magnitude $M$ are computed as follows. First, the CDFs of the prediction probability are obtained for each predicted class $C$ considering each drift magnitude $M$. Let $F_{C,M}$ denote the CDF of the predicted class $C$ for drift magnitude $M$. $F_{C,M}$ is a discrete function of the prediction

probability $p$ and it indicates the percentage of predictions with prediction probability below $p$. Let $G_{C,M}$ denote the gradient of the $F_{C,M}$ at each prediction probability $p$ with respect to the magnitude $M$. For a predicted class $C$ and drift magnitude $M$, $G_{C,M}$ is computed using the standard second-order approximation as,

$$G_{C,M} = (F_{C,M+1} - F_{C,M-1})/2. \tag{2}$$

For boundary magnitudes, $F_{C,M}$ and either the $F_{C,M-1}$ or the $F_{C,M+1}$ are used to compute the $G_{C,M}$. The threshold $\tau_{C,M}$ corresponds to the probability where the $G_{C,M}$ is maximum. For some drift type $T$, the set of thresholds $\tau_{C,M}$ for $1 \leq C \leq n$ and $0 \leq M \leq m$ is referred to as the threshold dictionary. The same procedure is followed to construct the dictionary of threshold for every potential drift type.

After the thresholds are computed, the next step is the application of the thresholds to estimate the magnitude. When a static class distribution is assumed, the drift magnitude for some drift type $T$ is estimated by comparing the observed percentage of predictions with the expected percentage of predictions above each threshold $\tau_{C,M}$, as described in the following. Let $P_{C,M}$ denote the expected percentage of predictions above $\tau_{C,M}$ and $\hat{P}_{C,M}$ denote the observed percentage of predictions above $\tau_{C,M}$ for the predicted class $C$ at drift magnitude $M$. A collection of expected percentages $P_{C,M}$ for $1 \leq C \leq n$ and $0 \leq M \leq m$, referred to as the percentage dictionary, is computed a priori. During the operation time, the observed percentages $\hat{P}_{C,M}$ are computed for $1 \leq C \leq n$ and $0 \leq M \leq m$ for a given batch of images. Thereafter, the absolute sum of the percentage errors $e_M = \sum_{i=1}^{C} \|P_{C,M} - \hat{P}_{C,M}\|$ is computed for each magnitude $M$. The estimated magnitude $M_e$, corresponds to the magnitude that has the minimum $e_M$.

As was mentioned in Section 2, the number of images predicted as a particular class and the prediction probability distribution of a class depends on the class distribution in the input data stream. Equation (1) indicates that the expected percentage $P_{i,M}$ above the threshold $\tau_{i,M}$ for the predicted class $C = i$ also depends on the number of images from each class i.e. the class distribution. When the class distribution changes, the expected percentage $P_{C,M}$ and the observed percentage $\hat{P}_{C,M}$ are not comparable. To address this challenge, we apply quantification to estimate the number of images from each class.

The proposed quantification method is a modified version of the *readme* method by Hopkins & King (2010). The *readme* method estimates the class distribution using the number of samples observed in each predicted class. In contrast, the proposed quantification method estimates the class distribution using the number of samples observed in each predicted class before and after applying the thresholds. The method considers the relation between the number of images predicted as a particular class and the number of images from each class, before and after thresholds are applied. The drift magnitude is estimated based on the quantification output.

The relation between the number of images predicted as class $C$ and the number of images from each class $\hat{C}$ is as follows. Let $y_i$ denote the number of images predicted as class $i$, $1 \leq i \leq n$. The quantity $y_i$ depends on the probability $P(C = i|\hat{C} = j)$ and the number of images from each class $x_j$, $1 \leq i, j \leq n$. The relation between the $y_i$, and the probability $P(C = i|\hat{C} = j)$, and $x_j$ is shown in Equation (3) below.

$$y_i = \sum_{j=1}^{n} P(C = i|\hat{C} = j)x_j. \tag{3}$$

The conditional probability $P(C = i|\hat{C} = j)$ depends on the model behavior and the dataset. $P(C = i|\hat{C} = j)$ is calculated by obtaining the confusion matrix of the classifier and normalizing the confusion matrix with the number of images from each class. The matrix of $P(C = i|\hat{C} = j)$ for $1 \leq i, j \leq n$ is referred to as the coefficient matrix and is denoted as $A_1$, which is of order $n \times n$.

Equation (3) is modified as below to obtain the relation between the number of images above the threshold $\tau_{C,M}$ in each predicted class $C$ and the number of images from each class $x_j$. Let $\widetilde{y}_i$ denote the number of images predicted as class $i$ with prediction probability $p \geq \tau_{i,M}$. Let $P((C = i \cap p \geq \tau_{i,M})|\hat{C} = j)$ denote the probability of being predicted as class $i$ such that the prediction probability $p$ is greater than or equal to the threshold $\tau_{i,M}$ given that the actual class is $j$. The $\widetilde{y}_i$ and

$x_j$ for $1 \leq i, j \leq n$ are related as in Equation (4).

$$\widetilde{y}_i = \sum_{j=1}^{n} P((C = i \cap p \geq \tau_{i,M})|\hat{C} = j)x_j. \tag{4}$$

To compute the probability $P((C = i \cap p_i \geq \tau_{i,M})|\hat{C} = j)$ for $1 \leq i, j \leq n$ for a given drift magnitude $M$, first, the confusion matrix is obtained after applying the thresholds $\tau_{i,M}$ for each predicted class $C$. Thereafter, the thresholded confusion matrix is normalized by the number of images from each class $\hat{C}$. The matrix of $P((C = i \cap p \geq \tau_{i,M})|\hat{C} = j)$ for $1 \leq i, j \leq n$ is referred to as the thresholded-coefficient matrix and is denoted as $A_2$, which is of order $n \times n$.

A system of linear equations is derived by considering $y_i$ and $\widetilde{y}_i$, $1 \leq i \leq n$ and $x_j$, $1 \leq j \leq n$. Let $Y = \{y_1, y_2, \ldots, y_n, \widetilde{y}_1, \widetilde{y}_2, \ldots, \widetilde{y}_n\}^T$, $X = \{x_1, x_2, \ldots, x_n\}^T$, and $A = \begin{bmatrix} A_1 \\ A_2 \end{bmatrix}$. Then,

$$Y = AX. \tag{5}$$

The number of images from each class $X$ is estimated by solving the linear equation system in Equation (5) using the least-squares method. Let $\hat{X}$ be the least-squares solution of the linear equation system in Equation (5). The residual of the solution is obtained by computing the Euclidean norm of the difference between $Y$ and $A\hat{X}$. The residual is an indication of how well the solution satisfies the linear equation system. Let $r_M$ denote the residual of the solution for magnitude $M$, i.e.,

$$r_M = Y - A\hat{X}. \tag{6}$$

For each drift magnitude, $X$ is estimated by substituting the $A$ in Equation (5) with the matrix corresponding to the magnitude $M$. The residual $r_M$ is computed for $0 \leq M \leq m$. The estimated drift magnitude $M_e$ is the magnitude with the minimum residual value,

$$M_e = argmin(r_0, r_1, \ldots, r_m). \tag{7}$$

For an accurate estimation of the drift magnitude, the drift type needs to be correctly identified when there is more than one potential drift type. First, we apply the magnitude estimation considering every potential drift type. If a majority of the drift types indicate a non-zero magnitude, we use a structured approach to detect the drift type. The approach relies on a classification neural network trained to detect the drift type, referred to as the type detection network, and the minimum residual value of the magnitude estimation in each drift type. The type detection network classifies every image in the input stream into one of the $N$ potential drift types. It is trained to identify the drift type present in an image, regardless of the drift magnitude. For better accuracy, it is trained using only the drifted images without including the clean images. Therefore, the type detection network output is used only when the magnitude estimation indicates a non-zero drift magnitude.

The complete flow of the proposed drift detection framework is as follows. The first step is to determine if a data drift has occurred in the input image stream. For a given batch of images, it is determined as a data drift has occurred if more than $\alpha$ types indicate a non-zero magnitude after applying the magnitude estimation. If a data drift is present, a scoring criterion is used to identify the drift type, and the drift type with the highest score is selected. The score is computed based the output of the type detection network and the minimum residual of each type. The percentage of the images detected as a particular drift type by the type detection network is used as one score value. Let $s_{t,T}$ denote the type detection network percentage for type $T$. The minimum residual value of each drift type is normalized with respect to the lowest residual value among all drift types. The percentage above the lowest residual is added as a negative score. $s_{r,T}$ denote the residual score for type $T$. The total score $s_T$ of the drift type $T$ is the summation of $s_{t,T}$ and $s_{r,T}$, i.e., $s_T = s_{t,T} + s_{r,T}$. The drift type corresponds to the type with the highest score and the magnitude corresponds to the magnitude with the minimum residual value of the detected drift type. See also Algorithm 1 in the Appendix.

## 4 EXPERIMENTAL RESULTS

The proposed drift detection framework was validated by considering the data drifts that commonly occur in image data streams due to various types of effects. We experimented with three noise

effects, namely Gaussian, Poisson, and Salt & Pepper, and three weather effects, namely Snow, Fog, and Rain. Experimental results are presented considering the six drift types on the MNIST, CIFAR10, and CIFAR100 datasets. Clean images without any data drift and images of different drift magnitudes were considered for each type.

For the MNIST dataset, we used a lightweight CNN that resembles the VGG architecture by (Simonyan & Zisserman (2014)). The network consisted of two convolutional layers, followed by a maxpooling layer, followed by a fully connected layer, followed by the output layer. It was trained using 60,000 images. For the CIFAR10 dataset, we use the ResNet18 architecture (He et al. (2016)). For the CIFAR100 dataset, we use the network was trained using 50,000 images. The ResNet50 architecture (He et al. (2016)). The network was trained using 60,000 images. CIFAR100 validation dataset only has 100 images per-class. Such a small number of samples does not provide sufficient statistical confidence, resulting in instability in the linear equation system. Therefore, a shadow network of the primary classifier was trained on the twenty super-classes of the CIFAR100 dataset, and drift detection was performed using the shadow network output. The accuracy of the MNIST, CIFAR10, and CIFAR100 (shadow network) classifiers were 99.1%, 86.7%, and 90.4% respectively.

The validation datasets consisting of 10,000 images in each were used in the drift magnitude estimations. 60% of the validation dataset was used to compute the thresholds ($\tau_{C,M}$), expected percentages above the thresholds ($P_{C,M}$), and the coefficient matrices ($A_1$ and $A_2$). Magnitude estimations were done using the remaining 40%. The architecture of the type detection network was the same as the architecture of the image classification network of each dataset. They were trained with transfer learning using the weights of the primary classification network.

The Drift magnitude of each noise effect was represented by the corresponding model parameter of the noise type as follows. In Gaussian with $\sigma^2$, in Poisson with $1/\lambda$, and in Salt & Pepper with $\sigma^2$. Figure A.4 shows an example MNIST image affected by the noise effects compared with the original clean image.

The drift magnitude of each weather effect was represented using the values within the range $[0, 1]$. Weather effects were added to the images using the Albumentations library Buslaev et al. (2020). Figure A.5 shows an example CIFAR10 image affected by the weather effects compared with the original clean image.

In the first experiment, a higher number of drift magnitudes were considered for each drift type within a selected range. In the second experiment, a lesser number of drift magnitudes compared to the first experiment were considered for each drift type within the same range.

The magnitude estimation method was evaluated considering twenty different random class distributions for each drift magnitude. A random class distribution was created as follows for a given magnitude. A total of 400 images from each class was considered, and a random percentage of the total images in each class was chosen to create a batch of images with a random class distribution. twenty such random distributions were considered for each magnitude. The class distribution of a single batch of images was skewed, with respect to the class distribution of the dataset used to compute the thresholds. The method was tested with a low-skew in the class distribution and with a high-skew in the class distribution. The skewness of the class distribution was changed by varying the percentage of images selected from each class. Note that the magnitude estimation results are presented considering the estimated magnitude by the method correspond to the correct drift type.

First we present the results pertaining to the drift detection and type detection by the proposed framework. Type detection depends on the accuracy of the type detection network and the residual of the solution of the linear equation system. We observed a high accuracy in the type detection network for all magnitudes of all drift types in all three datasets. The accuracy decreased slightly at lower magnitudes, the drift type of a batch of images was determined accurately. See also Figure A.8 in the Appendix. Furthermore, in order to accurately detect the drift type, the minimum residual value of the correct drift type should be lower than that of all the other types, and this was satisfied in majority of the cases.

Table 1 shows the drift detection accuracy and the type detection accuracy of the proposed framework observed in the first experiment in which a higher number of drift magnitudes were considered. 100% drift detection accuracy was observed in many drift types. In some drift types, drift detection

accuracy decreased because some of the lowest non-zero magnitudes were estimated as magnitude zero. The drift type was also detected accurately in most of the types. The type detection accuracy was slightly reduced for certain drift types because, on some occasions, the minimum residual of the actual type was higher than the minimum residual of some other drift type. Therefore, the score $s_T$ of the actual type was reduced.

| Dataset | Drift Type | Drift Detection Accuracy | Type Detection Accuracy |
|---|---|---|---|
| MNIST | Gaussian | 93.33% | 100% |
| | Poisson | 88.81% | 100% |
| | Salt & Pepper | 91.19% | 100% |
| CIFAR10 | Gaussian | 100% | 94.52% |
| | Poisson | 100% | 99.52% |
| | Salt & Pepper | 100% | 97.86% |
| | Snow | 100% | 100% |
| | Fog | 90.91% | 99.09% |
| | Rain | 100% | 100% |
| CIFAR100 | Gaussian | 100% | 100% |
| | Poisson | 100% | 100% |
| | Salt & Pepper | 100% | 100% |
| | Snow | 93.18% | 100% |
| | Fog | 90.00% | 100% |
| | Rain | 100% | 100% |

Table 1: Drift detection accuracy and the drift type detection accuracy of the proposed method.

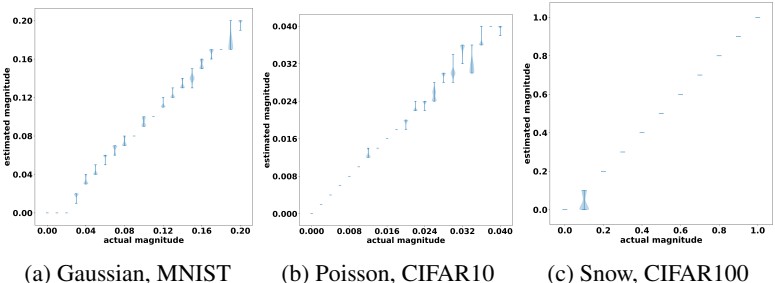

(a) Gaussian, MNIST      (b) Poisson, CIFAR10      (c) Snow, CIFAR100

Figure 2: Drift magnitude estimations of different drift types under low-skew class distributions.

Next, we present the drift magnitude estimation results of the first experiment. 2 shows magnitude estimations of the first experiment under low-skew class distributions. For each drift magnitude, the estimated magnitudes of the twenty different class distributions are shown in the form of a violin plot. The violin plot shows the distribution of the twenty estimations for a particular magnitude. Due to space limitations, figures are included only for one drift type in each dataset. It was observed that the quantization error of the estimations was within an acceptable range for all the drift types. Even though the estimated magnitude was not exact in some trials, the estimated magnitudes monotonically increased as the drift magnitude increased which prevented significant underestimations. Large underestimations are not desirable since they can compromise the reliability of the method. Furthermore, the small quantization error and monotonicity of the estimations allow the user to take appropriate remedial actions at a desired drift level even if the estimation is not exact.

Another observation was that, in some drift types, several lowest non-zero drift magnitudes were estimated as zero. This indicated the lowest drift magnitude of a drift type that the method can detect. For instance, in Figure 2, the three lowest magnitudes of Gaussian noise in the MNIST classifier, i.e., 0, 0.01, and 0.02, were estimated as zero. The lowest detected magnitude in the MNIST classifier of Gaussian, Poisson, and Salt & Pepper were 0.03, 0.3, and 0.06 respectively. Even though, some magnitudes were incorrectly estimated as zero, the impact of the data drift on the primary classification network was minimal at those magnitudes, and such imperfections were acceptable. Furthermore, the method always estimated a non-zero magnitude for every drift magnitude beyond a certain magnitude in all the drift types. For instance, in the magnitude estimations shown in Figure 2, the threshold magnitudes of Gaussian, Poisson, and Snow, beyond which the estimations were always non-zero, were 0.02, 0, and 0.1, respectively. This indicates that the proposed method has the ability to detect data drifts due to any type of effects that occur in images. It was observed that the method underestimated the drift magnitude for some drift types. For instance, drift magnitudes of Gaussian noise between 0.01-0.03 were always underestimated in Figure 2. Underestimations

were observed in several other drift types, but the quantization error was always within a acceptable range. See also the maximum quantization errors in Table 2.

A relatively higher quantization error and a variance in the magnitude estimations were observed under high-skew class distributions. Nevertheless, the quantization error was within an acceptable range. See also the maximum quantization errors in Table 2. The method was able to accurately distinguish among the magnitudes that were quite apart even under high-skew class distributions. See also Figure A.11 in the Appendix.

Next, we analyzed the quantization error of the proposed magnitude estimation method. For comparison purposes, we implemented the method by Senarathna et al. (2023) which we used as the baseline. Table 2 shows the quantization error summary of the magnitude estimations in the first experiment. The quantization error was analyzed considering low-skew class distributions and high-skew class distributions, and compared with the method by Senarathna et al. (2023). Quantization error was normalized by the quantization resolution for better interpretability. Quantization resolution is denoted by $\Delta$ and shown in the third column. The number of drift magnitudes $m$ is shown in the fourth column. For each drift type, the average, maximum, and standard deviation of the quantization error are shown in the table.

| | | | | Quantization Error (normalized) | | | | | | | | | | | |
|---|---|---|---|---|---|---|---|---|---|---|---|---|---|---|---|
| | | | | Low-Skew Class Distribution | | | | | | High-Skew Class Distribution | | | | | |
| Dataset | Drift Type | $m$ | $\Delta$ | Proposed | | | Senarathna et al. (2023) | | | Proposed | | | Senarathna et al. (2023) | | |
| | | | | Avg | Max | Std | Avg | Max | Std | Avg | Max | Std | Avg | Max | Std |
| MNIST | Gaussian | 20 | 0.01 | 0.77 | 2 | 0.35 | 0.89 | 3 | 0.40 | 0.96 | 5 | 0.77 | 1.26 | 7 | 1.64 |
| | Poisson | 20 | 0.1 | 1.64 | 3 | 0.42 | 1.50 | 3 | 0.40 | 1.52 | 4 | 0.56 | 1.19 | 8 | 1.30 |
| | Salt & Pepper | 20 | 0.02 | 0.48 | 3 | 0.37 | 0.86 | 3 | 0.52 | 0.64 | 5 | 0.86 | 1.08 | 6 | 1.41 |
| CIFAR10 | Gaussian | 20 | 0.001 | 0.45 | 3 | 0.39 | 1.43 | 7 | 2.51 | 0.62 | 5 | 0.65 | 2.32 | 14 | 5.89 |
| | Poisson | 20 | 0.002 | 0.32 | 2 | 0.35 | 1.30 | 9 | 2.75 | 0.54 | 3 | 0.57 | 2.40 | 15 | 6.88 |
| | Salt & Pepper | 20 | 0.002 | 0.33 | 2 | 0.25 | 0.59 | 3 | 0.56 | 0.47 | 3 | 0.38 | 2.25 | 14 | 4.04 |
| | Snow | 10 | 0.1 | 0.50 | 2 | 0.49 | 0.55 | 3 | 0.63 | 0.60 | 3 | 0.67 | 1.65 | 8 | 2.69 |
| | Fog | 10 | 0.1 | 0.20 | 2 | 0.18 | 0.22 | 3 | 0.23 | 0.37 | 3 | 0.39 | 1.00 | 4 | 1.05 |
| | Rain | 10 | 0.1 | 0.40 | 2 | 0.32 | 0.43 | 2 | 0.35 | 0.59 | 3 | 0.46 | 1.40 | 8 | 2.69 |
| CIFAR100 | Gaussian | 20 | 0.001 | 1.01 | 5 | 1.88 | 2.18 | 8 | 4.70 | 1.30 | 10 | 1.72 | 3.38 | 13 | 2.90 |
| | Poisson | 20 | 0.002 | 0.76 | 9 | 1.00 | 1.90 | 10 | 4.41 | 1.33 | 10 | 1.63 | 3.6 | 13 | 3.24 |
| | Salt & Pepper | 20 | 0.002 | 0.20 | 3 | 0.23 | 1.27 | 8 | 1.98 | 0.38 | 3 | 0.69 | 3.00 | 14 | 2.73 |
| | Snow | 10 | 0.1 | 0.06 | 1 | 0.06 | 0.47 | 2 | 0.29 | 0.60 | 3 | 0.61 | 1.17 | 6 | 2.05 |
| | Fog | 10 | 0.1 | 0.20 | 1 | 0.16 | 0.25 | 1 | 0.19 | 0.22 | 3 | 0.25 | 0.86 | 4 | 0.97 |
| | Rain | 8 | 0.1 | 0.49 | 2 | 0.45 | 0.57 | 2 | 0.44 | 0.50 | 2 | 0.32 | 0.76 | 4 | 0.58 |

Table 2: Quantization error summary of the drift magnitude estimations method compared with the method by Senarathna et al. (2023). A higher number of quantization levels are considered in this experiment. Error metrics are normalized by quantization resolution.

The average normalized quantization error was less than 1 for a majority of the drift types. This indicates that the method estimates the magnitude accurately most of the time. The average was between 1-2 in some drift types due to under-estimations happening at every magnitude. Even though there were under estimations in some drift types, the quantization error was within an acceptable range as reflected by the maximum and the standard deviation of those types. When compared to the method proposed by Senarathna et al. (2023), the proposed method always had equal or less average, maximum, and standard deviation except for the Poisson type in the MNIST dataset under low-skew class distributions. Especially, when the class distribution skew was high, the proposed method achieved a significant improvement over the method by Senarathna et al. (2023). This emphasizes the significance of the proposed quantification-based magnitude estimation criterion, which has the ability to cope with varying class distributions.

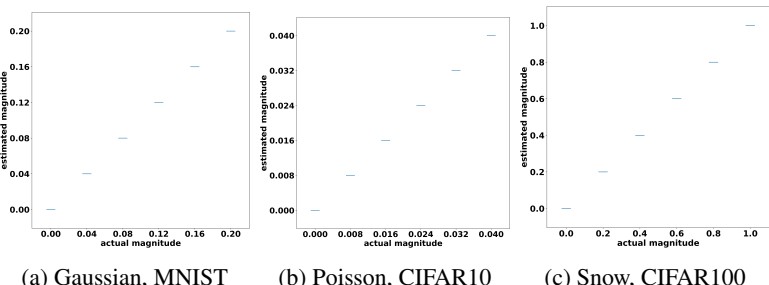

(a) Gaussian, MNIST     (b) Poisson, CIFAR10     (c) Snow, CIFAR100

Figure 3: Drift magnitude estimations of different drift types under low-skew class distributions.

In the second experiment, the method was evaluated with a lesser number of quantization levels. This experiment illustrates the accuracy of the method when the number of drift magnitudes is less and consecutive drift magnitudes are further apart from each other. Only five drift magnitudes within the same magnitude range as in the first experiment were considered and twenty different random class distributions were considered for each magnitude.

Estimation results of three drift types are shown in Figure 3 under low-skew class distributions. See also Figure A.12 in the Appendix. It was observed that the method was able to estimate the magnitudes with higher accuracy when the number of quantization levels was less.

Table 3 shows the quantization error summary of the second experiment. The average normalized quantization error was less than 1 for every drift types when a lesser number of quantization levels was considered. For most of the drift types the average was zero i.e. all the magnitudes were exactly estimated by the proposed method. The maximum normalized quantization error was 2 for any drift type. In comparison to the quantization errors of the first experiment, a significantly lesser average, maximum, and variance were observed in all the drift types.

| Dataset | Drift Type | $m$ | $\Delta$ | Quantization Error (normalized) | | | | | |
| | | | | Low-Skew Class Distribution | | | High-Skew Class Distribution | | |
| | | | | Avg | Max | Std | Avg | Max | Std |
|---|---|---|---|---|---|---|---|---|---|
| MNIST | Gaussian | 5 | 0.04 | 0 | 0 | 0 | 0.03 | 1 | 0.18 |
| | Poisson | 5 | 0.4 | 0.59 | 1 | 0.49 | 0.44 | 1 | 0.5 |
| | Salt & Pepper | 5 | 0.08 | 0 | 0 | 0 | 0.08 | 1 | 0.26 |
| CIFAR10 | Gaussian | 5 | 0.004 | 0 | 0 | 0 | 0.02 | 1 | 0.13 |
| | Poisson | 5 | 0.008 | 0 | 0 | 0 | 0 | 0 | 0 |
| | Salt & Pepper | 5 | 0.008 | 0 | 0 | 0 | 0 | 0 | 0 |
| | Snow | 5 | 0.2 | 0.18 | 2 | 0.58 | 0.35 | 2 | 0.76 |
| | Fog | 5 | 0.2 | 0 | 0 | 0 | 0 | 0 | 0 |
| | Rain | 5 | 0.2 | 0 | 0 | 0 | 0 | 0 | 0 |
| CIFAR100 | Gaussian | 5 | 0.004 | 0.07 | 1 | 0.25 | 0.14 | 2 | 0.37 |
| | Poisson | 5 | 0.008 | 0.15 | 1 | 0.36 | 0.21 | 2 | 0.43 |
| | Salt & Pepper | 5 | 0.004 | 0 | 0 | 0 | 0 | 0 | 0 |
| | Snow | 5 | 0.2 | 0 | 0 | 0 | 0 | 0 | 0 |
| | Fog | 5 | 0.2 | 0 | 0 | 0 | 0.02 | 1 | 0.13 |
| | Rain | 4 | 0.2 | 0 | 0 | 0 | 0 | 0 | 0 |

Table 3: Quantization error summary of the proposed drift magnitude estimations method with a lesser number of quantization levels. Error metrics are normalized by quantization resolution.

## 5 CONCLUSION

A novel framework is proposed in this paper for detecting data drifts occur in the input stream of image classification neural networks due to various effects. The framework can detect data drifts, identify the type of drift, and estimate the drift magnitude in occur in image data streams. It is consisting of a classification network that detects the drift type and a novel statistical method that estimates the drift magnitude. The magnitude estimation method relies on the change in the prediction probability distribution caused by the data drift. The drift magnitude is estimated by applying a set of thresholds to the prediction probabilities. Thresholds are applied based on the predicted class, and the magnitude is estimated using the percentage of predictions above the threshold in each class. Experimental evaluation was conducted considering three different classification neural networks trained on MNIST, CIFAR10, and CIFAR100 datasets. Data drifts of different magnitudes that occur in images caused by various types of noise effects and weather effects were considered in the evaluation. Results indicated that the proposed framework can detect data drifts with a high accuracy, while accurately identifying the drift type and estimating the drift magnitude with a very low quantization error.

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

## A APPENDIX

### A.1 IMPACT OF DATA DRIFT ON THE ACCURACY OF THE MNIST CLASSIFIER

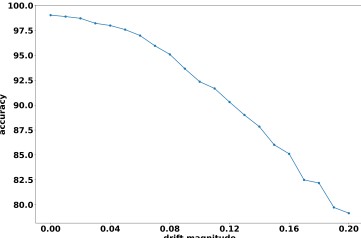

Figure A.1: Impact of data drift due to Gaussian noise on the MNIST classifier.

## A.2 PREDICTION PROBABILITY DISTRIBUTION OF THE MNIST CLASSIFIER

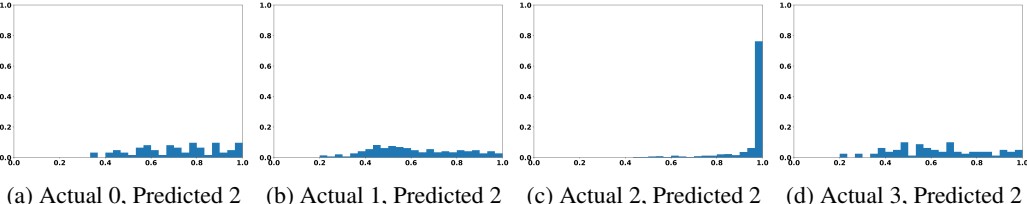

(a) Actual 0, Predicted 2    (b) Actual 1, Predicted 2    (c) Actual 2, Predicted 2    (d) Actual 3, Predicted 2

Figure A.2: Prediction probability distribution of the images from different classes predicted as class 2 in the MNIST classifier with $\sigma^2 = 0.20$ Gaussian noise.

Consider the images predicted as class 2 in the MNIST classifier under $\sigma^2 = 0.20$ Gaussian noise. Figure A.2 shows the distributions of the prediction probabilities of the images predicted as class 2, belonging to four different classes $0, 1, 2, 3$. It can be observed that the distribution of the prediction probabilities of the images that are predicted as class 2 is different for the images belonging to different classes. Similarly, the images from the other classes predicted as class 2 had different distributions. Therefore, the prediction probability distribution of the images predicted as class 2 depends on the class distribution.

## A.3 THRESHOLD DICTIONARY

An example threshold dictionary in shown in Table A.1 considering Gaussian noise in the MNIST classifier. The classifier consists of classes 0 through 9 (the columns of the array), and six drift magnitudes (the rows of the array) were considered. CDFs in Figure 1a correspond to $F_{1,M}$ and CDFs in Figure 1b correspond to $F_{(2,M)}$ of six magnitudes where $0 \leq M \leq 0.20$. The $\tau_{1,M}$ values in the third column of Table A.1 are calculated using the $F_{1,M}$ shown in Figure 1a, and the $\tau_{2,M}$ values in the fourth column of Table A.1 are calculated using the $F_{2,M}$ shown in Figure 1b. Similarly, the thresholds for all the classes for $0 \leq C \leq 9$ in Table A.1 are computed.

| Magnitude | Class | | | | | | | | | |
|---|---|---|---|---|---|---|---|---|---|---|
| | 0 | 1 | 2 | 3 | 4 | 5 | 6 | 7 | 8 | 9 |
| $\sigma^2 = 0$ | 0.98 | 0.98 | 0.98 | 0.98 | 0.98 | 0.98 | 0.98 | 0.98 | 0.98 | 0.98 |
| $\sigma^2 = 0.04$ | 0.98 | 0.98 | 0.98 | 0.98 | 0.98 | 0.98 | 0.98 | 0.98 | 0.98 | 0.98 |
| $\sigma^2 = 0.08$ | 0.98 | 0.98 | 0.98 | 0.98 | 0.98 | 0.98 | 0.98 | 0.98 | 0.98 | 0.98 |
| $\sigma^2 = 0.12$ | 0.98 | 0.94 | 0.98 | 0.98 | 0.98 | 0.98 | 0.98 | 0.96 | 0.98 | 0.89 |
| $\sigma^2 = 0.16$ | 0.98 | 0.71 | 0.98 | 0.96 | 0.95 | 0.97 | 0.97 | 0.91 | 0.98 | 0.76 |
| $\sigma^2 = 0.20$ | 0.98 | 0.98 | 0.98 | 0.98 | 0.98 | 0.98 | 0.98 | 0.98 | 0.98 | 0.98 |

Table A.1: The threshold dictionary of the MNIST image classifier considering five different magnitudes of Gaussian noise.

## A.4 SAMPLE COEFFICIENT MATRICES

The matrix of $P(C = i | \hat{C} = j)$, for $1 \leq i, j \leq n$, shown in Figure A.3a was obtained by normalizing each column in the confusion matrix with the total number of images from the corresponding class $\hat{C}$. Figure A.3b shows the matrix of $P((C = i \cap p \geq \tau_{i,M}) | \hat{C} = j)x_j$ correspond to the matrix in Figure A.3a under no data drift in the MNIST classifier.

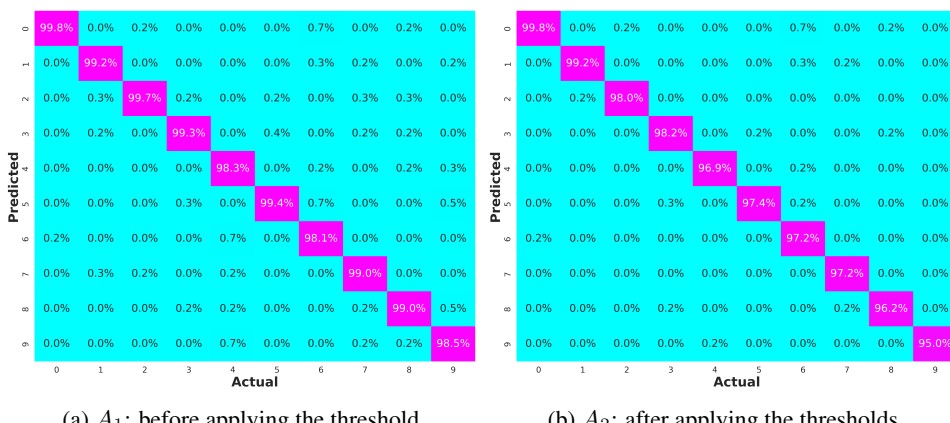

(a) $A_1$: before applying the threshold  (b) $A_2$: after applying the thresholds

Figure A.3: The probability of being predicted as class $C = i$ for the images from class $\hat{C} = j$ in the MNIST classifier under no data drift.

### A.5   PSEUDO CODE OF THE PROPOSED DRIFT DETECTION ALGORITHM

The complete flow of the proposed drift detection framework is shown in Algorithm 1. First, the algorithm determines if a data drift has occurred in the input. For a given batch of images $\mathbb{I}$, it is determined that a data drift has occurred if more than $\alpha$ types indicate a non-zero magnitude after applying the magnitude estimation. $\alpha$ is set to the half of the number drift types. If a data drift is detected, the type and the magnitude is determined using the criterion explained in Section 3.

---

**Algorithm 1** Drift_Detection($\mathbb{I}$)

---

1: **Input:** $\mathbb{I}$
2: **Output:** $type, M_e$
3: $total \leftarrow \text{length}(\mathbb{I})$
4: $min\_residual \leftarrow \{\}$
5: $magnitude \leftarrow \{\}$
6: $score \leftarrow \{\}$
7: $count \leftarrow 0$
8: **for** each type $N$ **do**
9: $\quad (r_{M,N}, M_{e,N}) \leftarrow \text{estimate\_magnitude}(\mathbb{I}, N)$
10: $\quad min\_residual \leftarrow min\_residual \cup \{r_{M,N}\}$
11: $\quad magnitude \leftarrow magnitude \cup \{M_{e,N}\}$
12: $\quad$ **if** $M_{e,N} > 0$ **then**
13: $\quad\quad count \leftarrow count + 1$
14: $\quad$ **end if**
15: **end for**
16: **if** $count \geq \alpha$ **then**
17: $\quad type\_array \leftarrow \text{type\_detection}(\mathbb{I})$
18: $\quad r_{\min} \leftarrow \min(min\_residual)$
19: $\quad$ **for** each type $N$ **do**
20: $\quad\quad c \leftarrow \text{count\_occurrence}(type\_array, N)$
21: $\quad\quad s_{t,T} \leftarrow \frac{c \times 100}{total}$
22: $\quad\quad s_{r,T} \leftarrow 100 - \frac{min\_residual[N] \times 100}{r_{\min}}$
23: $\quad\quad s_T \leftarrow s_{t,T} + s_{r,T}$
24: $\quad\quad score \leftarrow score \cup \{s_T\}$
25: $\quad$ **end for**
26: $\quad type \leftarrow \text{argmin}(score)$
27: $\quad M_e \leftarrow magnitude[type]$
28: **else**
29: $\quad type \leftarrow \text{None}$
30: $\quad M_e \leftarrow 0$
31: **end if**
32: **return** $(type, M_e)$

---

### A.6 SAMPLE IMAGES IMPACTED BY NOISE AND WEATHER EFFECTS

Figure A.4 shows an example MNIST image affected by the noise effects compared with the original clean image. Figure A.5 shows an example CIFAR10 image affected by the weather effects compared with the original clean image.

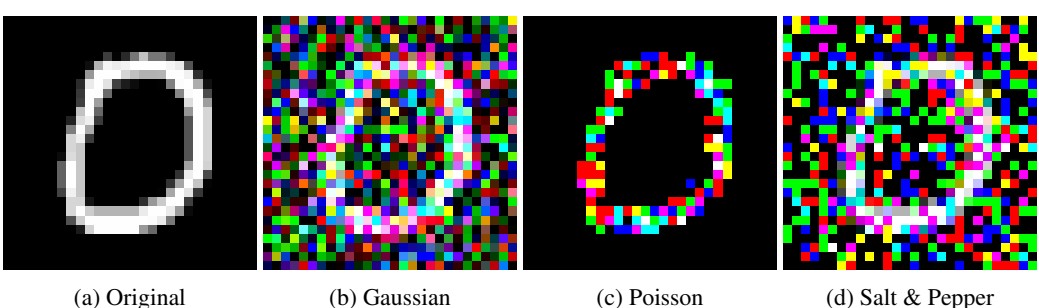

(a) Original        (b) Gaussian        (c) Poisson        (d) Salt & Pepper

Figure A.4: Noise effects on the MNIST data.

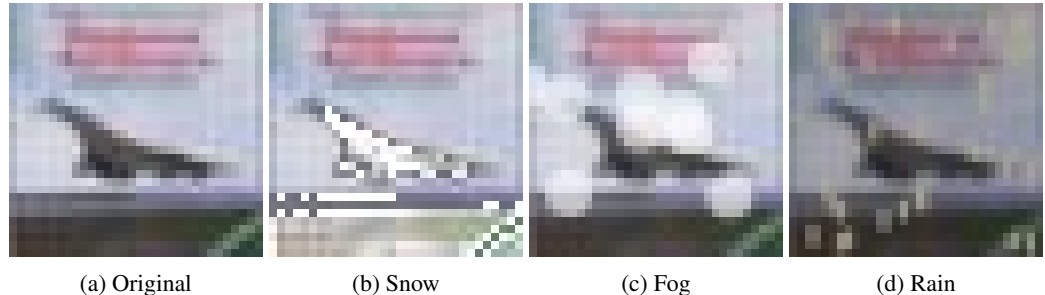

(a) Original      (b) Snow      (c) Fog      (d) Rain

Figure A.5: Weather effects on the CIFAR10 data.

## A.7 DRIFT MAGNITUDE ESTIMATION UNDER STATIC CLASS DISTRIBUTION

We assessed the quality of the thresholds computed by the proposed method by applying the thresholds to estimate the drift magnitude of a dataset with a static class distribution. In this experiment, the class distribution of the dataset used to compute the thresholds and the dataset used to estimate the drift magnitude were the same. Magnitude estimation was done using the criterion explained in Section III for the static class distribution scenario. Figure A.6 shows the estimated magnitudes of three drift types considering the three datasets. Blue dots show the estimated magnitude $M_e$ by applying the thresholds and the dotted line shows the ideal $M$ vs $M_e$ curve. Estimation results indicated that the thresholds computed by the proposed method can distinguish among different drift magnitudes with a high accuracy.

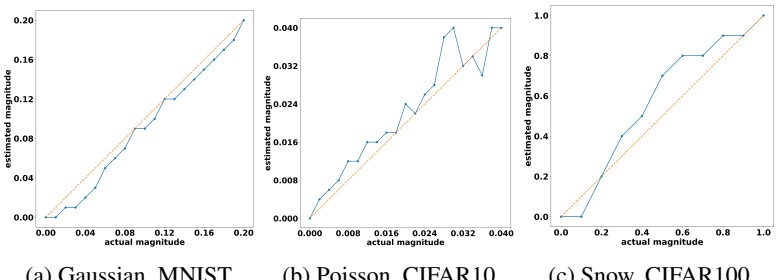

(a) Gaussian, MNIST      (b) Poisson, CIFAR10      (c) Snow, CIFAR100

Figure A.6: Drift magnitude estimation results under a static class distribution in the dataset.

## A.8 RESIDUAL PLOTS

The proposed magnitude estimation method relies on the residual of the linear equation system $r_M$ of each magnitude M. For the method to be accurate, the minimum $r_M$ should ideally be at the actual magnitude M. Figure A.7 shows several residual plots ($M$ vs $r_M$) of the method for Gaussian noise, considering four different drift magnitudes. Residual plots were obtained considering a single trial for each magnitude of the two drift types. It was observed that the error graph always had a convex region around the actual drift magnitude such that the minimum is equal or very close to the actual magnitude. This observation supports the validity, consistency, and robustness of the proposed method.

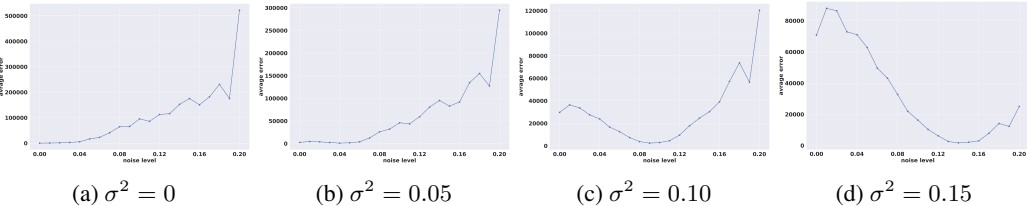

(a) $\sigma^2 = 0$       (b) $\sigma^2 = 0.05$       (c) $\sigma^2 = 0.10$       (d) $\sigma^2 = 0.15$

Figure A.7: Residual ($r_M$) plot of Gaussian noise on the MNIST classifier for five different drift magnitude estimations.

### A.9 TYPE DETECTION ACCURACY

The type detection network accuracy of the CIFAR10 dataset at different drift magnitudes of Gaussian noise is shown in Figure A.8. Figure A.9 shows the minimum residual value computed by the method for each drift type during magnitude estimations. Results are shown for twenty magnitudes of Gaussian noise in the CIFAR10 classifier considering twenty different class distributions for each magnitude. In most of the experiments, the minimum residual value of the Gaussian type was smaller than that of the other types. In other experiments, although the minimum residual value of Gaussian was not the lowest, it was close to being the lowest.

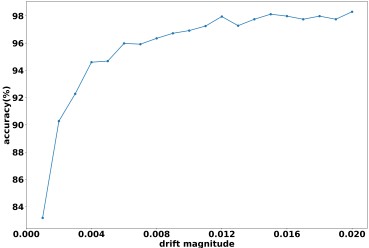

Figure A.8: Type detection network accuracy for the CIFAR10 dataset at different drift magnitudes of Gaussian noise.

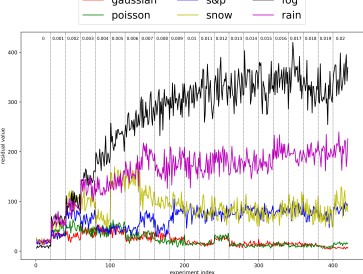

Figure A.9: Minimum residual value computed by the method for each drift type during magnitude estimations. Results are shown considering twenty different class distributions for each magnitude of Gaussian noise in the CIFAR10 classifier.

### A.10 CORRELATION BETWEEN THE QUANTIFICATION ERROR AND THE QUANTIZATION ERROR

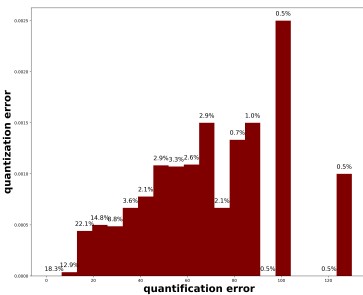

Figure A.10: The average quantization error for different ranges of the quantification error considering Gaussian noise in the CIFAR10 classifier. Percentages shown in the figure indicate the probability of quantification error being within that range.

We analyzed the correlation between the quantification error and the quantization error. In all the drift types, it was observed that the quantization error remained consistently low when the quantification error was at its lowest. Specifically, in all drift types, the average quantization error was minimal within the lowest quantification error range. This observation indicates the significance of the quantification for the proposed method to accurately estimate the drift magnitude. Figure A.10 shows the average quantization for different ranges of the quantification error considering Gaussian noise in the CIFAR10 classifier.

### A.11 EXPERIMENT 1: DRIFT MAGNITUDE ESTIMATION WITH A HIGHER NUMBER OF QUANTIZATION LEVELS UNDER VARYING CLASS DISTRIBUTION

Figure A.11 shows the magnitude estimation results of the first experiment under high-skew class distributions corresponding to the same drift types as those shown in Figure 2.

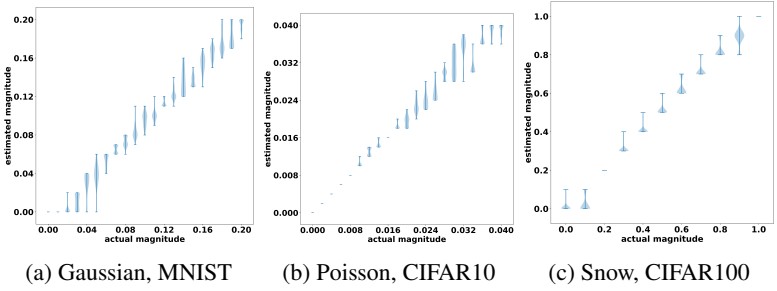

(a) Gaussian, MNIST          (b) Poisson, CIFAR10          (c) Snow, CIFAR100

Figure A.11: Drift magnitude estimations of different drift types under high-skew class distributions.

### A.12 EXPERIMENT 2: DRIFT MAGNITUDE ESTIMATION WITH A FEWER NUMBER OF QUANTIZATION LEVELS UNDER VARYING CLASS DISTRIBUTION

Figure A.12 shows the magnitude estimation results of the first experiment under high-skew class distributions corresponding to the same drift types as those shown in Figure 3.

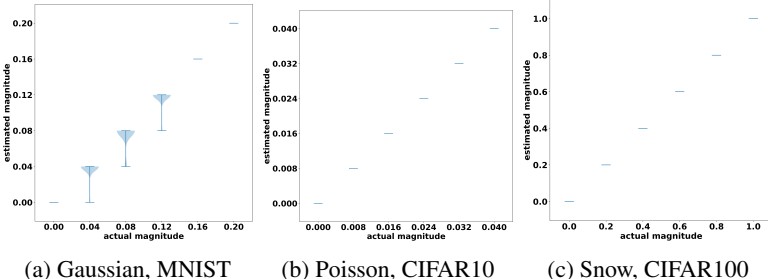

(a) Gaussian, MNIST     (b) Poisson, CIFAR10     (c) Snow, CIFAR100

Figure A.12: Drift magnitude estimations of different drift types under high-skew class distributions.

### A.13 DRIFT MAGNITUDE ESTIMATION IN A REAL APPLICATION SIMULATION

At last, we present the results of an experiment that illustrates the behavior of the proposed method in a real application environment. For this experiment, we created a dataset by stacking images from different drift magnitudes in the ascending order of the magnitude. Thereafter, we applied the magnitude estimation on a moving window of size 1000 with a step size of 200. An equal number of images were included for every drift level and they were ordered randomly such that the distribution among the classes varies as the window moves. Figure A.13 shows the magnitude estimation results obtained using the proposed method for Gaussian noise and Snow effect. The red line shows the estimated magnitude by the method and the dotted line shows the actual magnitude at the time. The drift magnitude of the majority was considered as the actual when the moving window overlapped with two magnitudes.

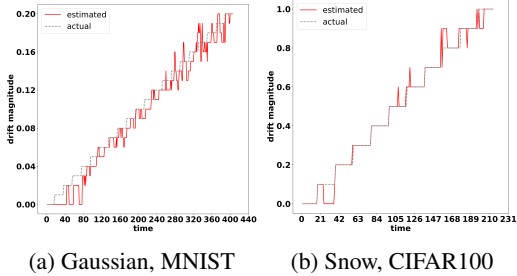

(a) Gaussian, MNIST     (b) Snow, CIFAR100

Figure A.13: Drift magnitude estimations by the proposed method in a simulation of a real application environment.

