# OpenReview forum: "Drift Type and Magnitude Detection in Image Classification Neural Networks"
_ICLR.cc/2025/Conference — ICLR 2025 Conference Withdrawn Submission_

### Official Review · Reviewer_zwwk · 2024-10-29

**Soundness:** 1
**Presentation:** 1
**Contribution:** 2
**Rating:** 3
**Confidence:** 4

**Summary:**

The paper proposes a framework to detect data distribution changes in image classification neural networks by monitoring changes in prediction probability distributions. It claims to identify both the type and magnitude of the distribution shift by combining a classification network (for identifying noise/weather effects) with a quantification-based method that estimates class proportions, though it confuses the concept of drift types with different noise patterns.

**Strengths:**

The paper attempts to provide a comprehensive solution by jointly considering drift detection, magnitude estimation, and handling varying class distributions in a single framework.

**Weaknesses:**

Weaknesses:

1. The first paragraph of the Introduction is entirely devoted to a general introduction of DNNs, without any mention of drift. Given that the paper's core focus is on detecting drift types and drift magnitude, I believe the DNN-related introduction is not central to this paper, and this entire paragraph provides little valuable information to readers.
2. The paper is poorly written. A paper should highlight its key points and quickly convey its innovations to readers. In the introduction, three paragraphs are spent on related work, while only one paragraph describes the paper's contribution, and even this paragraph fails to intuitively explain why the proposed method would work.
3. The first paragraph of Preliminaries is entirely about concept drift. So I assume the paper aims to address concept drift issues. If this is the case, there is a serious misuse of terminology. In concept drift, drift types include abrupt drift, gradual drift, recurrent drift, etc. While this paper uses the term "drift type" 86 times, it never explains or strictly defines what drift type means. According to Table 1 in the paper, the authors treat "gaussian noise, poisson noise, salt noise, snow fog rain, etc" as drift types. I find this inappropriate as these are more like different types of concepts. In summary, drift type is a specialized term in concept drift [1].
4. Line 163 states: "Pi,j denote the prediction probability distribution of the images belonging to the class j predicted as class i", but according to equation 1, I believe p_ij is a scalar, not a distribution. This appears to be an expression issue. I believe the paper consistently misuses the term "distribution".
5. Line 183, "per each drift type" should be changed to "for each effect type".
6. Lines 117-119 state: "To understand the impact of data drifts on image classification neural networks, let us consider the impact of Gaussian noise on a classification network trained on the MNIST handwritten digit image dataset, detailed in Section 4, under the effect of Gaussian noise." This sentence is highly redundant.
7. The experimental evaluation is inadequate as it only compares against a single baseline. For a paper proposing a new framework, comparing with multiple state-of-the-art methods is essential to demonstrate the effectiveness and advantages of the proposed approach. The limited comparison significantly weakens the paper's experimental validation.

Overall, I find the paper poorly written, with issues including misuse of terminology, redundant expressions, unclear logical flow, and lack of focus. Moreover, the paper seems to compare against only one baseline, which makes the experimental results unconvincing to me.


[1] Agrahari, S. and Singh, A.K., 2022. Concept drift detection in data stream mining: A literature review. Journal of King Saud University-Computer and Information Sciences, 34(10), pp.9523-9540.

**Questions:**

Same as weakness.

---

### Official Review · Reviewer_7nHb · 2024-10-29

**Soundness:** 2
**Presentation:** 2
**Contribution:** 2
**Rating:** 1
**Confidence:** 4

**Summary:**

The article proposes novel methods to detect drift, drift type, and drift magnitude in an image processing data set. The uniqueness of the approach seems to be its ability to deal with varying frequencies of the observed classes. The technical aspects are based on the CDFs of predicted classes and a usage of previous ideas proposed by Senarathna et al. The method is demonstrated on images which are distorted by different types of plausible noise, hence ground truth for evaluation is available in this case.
The article motivates the relevance of the research with a reference to deep vision models and the non-robustness w.r.t. noise.

**Strengths:**

The article deals with a relevant area given that models are delivered to the market and hence need to act in open environments. Further, it integrates an interesting SOTA.

**Weaknesses:**

The article faces the following weaknesses, in my opinion:
- The article fails to give valid mathematical definitions of relevant quantities, specifically what exactly is referred to by drift magnitude? Further, it shares the challenge of many other approaches which fail to give a definition of the underlying mathematical object (mathematically referred to as a drift process in several recent publications which aim for a formal definition of the statistical object dealt with)
- The article describes the novel approach within the text. This makes it very hard to catch the steps. A separate display of the algorithm would be appreciated.
- Properties of the algorithmic steps (including potential guarantees or deeper mathematical insight) are not set apart and not formalized, leaving a mere heuristic approach.
- The types of drift are a bit unclear, especially as regards the claim that every type of drift can be detected. This is not true as it works via the supervision and according classes, see e.g. the theoretical insight and proofs provided in https://www.scitepress.org/Papers/2023/117975/117975.pdf
- The relevance of the results is not clear. On the one hand, all data are theoretical one and no real observed drift is included. On the other hand, the benefit of detecting and specifically quantifying drift is unclear, as drift can have unpredictable consequences which do not necessarily depend on its magnitude.

**Questions:**

I suggest to include formal definitions of drift process, drift, drift magnitude.
I suggest to restructure the paper and set apart definitions, observations, algorithms, theorems (if any).
I would appreciate theoretical insight or statistical guarantees.
I would appreciate real data (e.g. Rialto Bridge is such a data set generated from a webcam).
Please be more precise w.r.t. detection of real / virtual drift and its relation to the model error, as this is nor mutually equivalent at all.
A specific downstream tasks which demonstrates the usability of the findings would be good.

---

### Official Review · Reviewer_URgh · 2024-11-03

**Soundness:** 2
**Presentation:** 3
**Contribution:** 2
**Rating:** 5
**Confidence:** 4

**Summary:**

This paper addresses issues of drift type and magnitude detection in the image classification task, which is an interesting topic. A learning method that relies on handling the changeable data has been proposed with ideal experimental results. However the learning setting and novelty are not well addressed. There are some suggestions for further improvement.

**Strengths:**

The topic of drift type and magnitude detection in the image classification task is interesting, and sufficient experiment has been given for model evaluation.

**Weaknesses:**

1. The problem definition is not clear, since the definition only reflects the probability change without considering the timestamp, which is an important term in data stream learning.
2. The threshold setting for drift magnitude identification still need more novelty.

**Questions:**

1.	In the data stream, concept drift means that the data distribution changes timely, which can be defined as $P(y_t | x_t) /neq P(y_{t+1} | x_{t+1})$, where $P(y|x)$ is the data distribution, $t$ and $t+1$ is the time point. However, I have not seen a clear definition of the problem in this paper.  Even in Eq. (1), the conditional probability has been defined, but the time point has not been expressed, this is a very important term in data stream learning. So, the author should further consider the problem definition.
2.	The drift type in this paper is simulated based on the Gaussian, Poisson, and so on. Actually, drift types in the data stream include sudden drift, incremental drift, gradual drift, and recurrent drift, which occur at different time points. So, it is necessary to give an explanation.
3.	Does the learning mode in this paper prequential test-then-train? Or offline training and online testing? If not, this paper handles the distribution change in image classification, not the data stream learning task.
4.	The threshold setting for drift magnitude identification seems a hyperparameter, a sensitivity analysis is required.
5.	It is needed a clear figure to show the proposed learning method.

---

### Official Review · Reviewer_DDRT · 2024-11-04

**Soundness:** 2
**Presentation:** 1
**Contribution:** 1
**Rating:** 1
**Confidence:** 2

**Summary:**

As the title specifies, the paper proposes a method for detecting drift in image data. It separates the process into the phases of detecting whether drift is present, classification of the type of drift, and estimation of the magnitude of the drift. They are certianly not the only researchers working on this.

**Strengths:**

I find it hard to see any strengths in this work. A not very well-known algorithm for drift detection in images has been improved. That is certainly positive.

**Weaknesses:**

Their only point of comparison is a method published by Senarathna et al. 2023 at a conference on high-performance switching and routing. The work improves upon the previous method by bringing in a method for estimating a class distribution that published in 2010 in a journal on political science. The choice of these non-standard techniques makes it hard to position this work in the context of the current state-of-the-art of drift detection, in particular as there is also no comparison to other, better known works in this area.

The paper is also not very precise in formulating its claims. The first section (2 pages!) is a very general introduction to concept drift detection that does not contribute much to a deeper understanding. This place could have been used by maybe bringing in some results from the appendix, which are frequently referenced and discussed in the main paper, but not part of the main paper. I judged the paper as a stand-alone contribution, and did not look at the appendix, which made it hard to follow the discussion.

The presented evaluation is rather straight-forward. Three standard datasets have been trained with standard neural network architectures, and drift was only simulated. The sole point of comparison is the above-mentioned work by Senarathna et al., with which I am not familiar (p.9, e.g., contains 7 references to that method, no reference to any other method). In any csae, just one comparison is not sufficiently convincing.

In addition, the paper is also not carefully written and/or proof-read. There are many very odd mistakes. For example, on p.7, after the three network architectures have been specified, there is an incomplete sentence "The ResNet50 architecture (He et al. (2016))." I am not sure what this refers to, probably it is the network that has been used to train on the CIFAR 100 dataset. Or the phrase "in occur" in the conclusions.

Minor comments:

  - "drift" does not have a plural, as far as I know.

  - "cumulative sum": Is there also a non-cumulative sum?

  - The authors use the symbol for set intersection (\cap) for denoting logical conjunction (\wedge). There is clearly a relation, but this choice is nevertheless confusing.

**Questions:**

I don't have any questions whose answers would have an impact on my evaluation.

---

### Note · Authors · 2025-02-17

I have read and agree with the venue's withdrawal policy on behalf of myself and my co-authors.

---

### Meta-Review · Area_Chair_wuCz · 2024-12-21

**Metareview:**

The paper concerns the problem of detecting data drifts in image classification. The introduced method is based on monitoring of changes in prediction probability distributions.

The Reviewers find the paper, among other issues, to be poorly written, lacking precise definitions of the concepts used, and referencing contributions that are not considered state-of-the-art.

All Reviewers rated the paper below the bar and the Authors did not prepare any rebuttal.

**Additional Comments On Reviewer Discussion:**

There was no discussion as the Authors did not sent rebuttal.

---

### Decision · Program_Chairs · 2025-01-22

Reject